# Fretting Wear Characteristics of SLM-Formed 316L Stainless Steel in Seawater

**Mingji Huang [1,2,\*], Ping Chen [1] and Xiaoxi Qiao [1]**

[1] School of Mechanical Engineering, University of Science and Technology Beijing, Beijing 100083, China
[2] Shunde Innovation School, University of Science and Technology Beijing, Foshan 528300, China
[\*] Correspondence: huangmingji@ustb.edu.cn

**Abstract:** The fretting wear characteristics of two different energy density 316L stainless steels formed by selective laser melting (SLM) under different friction conditions are studied. The image method was used to study the porosity of two samples with different energy densities (46.88 J/mm$^3$, 98.96 J/mm$^3$) formed by SLM. The dynamic wear test, respectively, evaluates its wear morphology and wear depth under three conditions: dry friction, distilled water, and an 3.5% NaCl solution. The porosity of the samples with SLM forming an energy density of 46.88 J/mm$^3$ and 98.96 J/mm$^3$ are 7.66% and 1.00%, respectively. Under the three conditions, the friction coefficient and wear depth of the samples with high energy density are smaller than those of the samples with low energy density; the friction of the samples with two energy densities in aqueous solution is faster than dry friction in air and tends to be stable. The friction coefficient in 3.5% NaCl solution is the smallest; when the energy density is constant, the wear depth of the fretting wear is the largest under dry friction and the smallest in distilled water. Under dry-friction conditions, the wear mechanisms of fretting wear are mainly oxidative wear and adhesive wear. In the fretting wear in the distilled water and the 3.5% NaCl solution, both wear mechanisms are abrasive wear and fatigue wear.

**Keywords:** selective laser melting; 316L stainless steel; energy density; porosity; fretting wear

## 1. Introduction

Selective laser melting (selective laser melting, SLM) is a technology that uses laser heat to quickly melt and solidify metal powder and generate a physical model layer by layer. It can produce very complex and personalized metal parts and has been widely used in multiple engineering fields [1–3]. Porosity is a common defect in the forming process of SLM. It usually reduces the density of the material and affects other related properties. Porosity is used to quantify the pore defects of the sample, such as pores, cracks, holes, etc. When producing parts through SLM technology, the first goal is to obtain close to full-density components. High porosity is not conducive to mechanical properties. Olakanmi et al. [4] stated that SLM is a very complex process and believed that process parameters would affect the densification mechanism and microstructure characteristics of SLM processed materials. Cherry et al. [5] studied the influence of laser energy density on SLM forming and summarized the optimal energy density.

The literature [6] shows that changing the scanning distance and energy density will directly affect the surface roughness, porosity, and friction characteristics of the formed sample. These documents show that changing the energy density can directly affect the performance of the formed sample.

The material properties are influenced by the forming process [7,8]. 316L stainless steel has good mechanical properties, but it is easy to wear [9,10]. Sun et al. [11] showed that the pore defects produced in the process of forming 316L stainless steel by SLM will lead to accelerated wear in the process of non-lubricating friction. The literature [12] compared the effects of three different processing techniques (selective laser melting, hot press sintering,

and conventional casting) on the microstructure and mechanical properties of 316L. The results show that SLM can improve its mechanical and wear properties in forming 316L. Huang et al. [13] studied the wear performance of SLM-formed 316L specimens and concluded that SLM-formed 316L has better wear performance than rolled 316L.

316L stainless steel is extensively used in marine engineering due to its superior corrosion resistance [14,15]. Seawater is a natural electrolyte and can damage the passive film of the metals, leading to corrosion, which will affect the tribological properties [16,17]. Meanwhile, small displacements can occur between close-fitting parts due to wind and wave effects, causing fretting wear. Several scholars have investigated the fretting wear characteristics of metallic materials in seawater [18,19], but less research exists on SLM-formed 316L.

In summary, the 3.5% NaCl solution is used to simulate seawater, and the fretting wear under dry friction and distilled water conditions is compared with the fretting wear under 3.5% NaCl conditions to study the performance of SLM-formed 316L stainless steel samples with different energy densities in seawater.

## 2. Materials and Methods

SLM molding uses 316L stainless steel powder. The particle size of the powder is shown in Figure 1, and its element composition is shown in Table 1. Selective laser melting adopts a metal 3D printer model EP-M100T (made in Eplus3D Corporation, Beijing, China) with a maximum power of 200 W. The machine is equipped with an IPGYt fiber laser (made in Eplus3D Corporation, Beijing, China) with a wavelength of 1030 nm and a spot diameter of 50 μm. All samples are in an atmosphere with an oxygen content of ≤1000 mg/L.

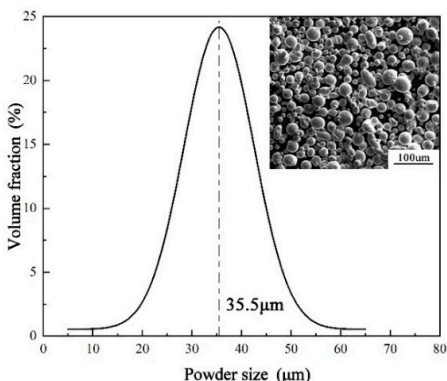

**Figure 1.** 316L powder micro-morphology and particle size distribution.

**Table 1.** Chemical composition of gas-atomized 316L powder (wt%).

| C | Si | Mn | S | P | Cr | Ni | Mo | Fe |
|---|---|---|---|---|---|---|---|---|
| ≤0.03 | ≤1.0 | ≤2.0 | ≤0.03 | ≤0.035 | 16.0–18.0 | 10.0–14.0 | 2.0–3.0 | Bal. |

The main process parameters in the SLM-forming process are laser power $L$ (W), scanning speed $v$ (mm/s), layer thickness $t$ (mm), and scanning distance $h$ (mm). According to formula (1) [20], these variables can be combined into energy density $E$ (J/mm$^3$). The literature [6] studied the effect of energy density on the porosity of 316L. The study showed that in the range of energy density of 50 J/mm$^3$~200 J/mm$^3$, the porosity first decreased and then increased with the increase in energy density, and when the energy density was 125 J/mm$^3$, the porosity reached the lowest value ( the lowest value was 0.63%). In this experiment, two samples with energy densities of 46.88 J/mm$^3$ and 98.96 J/mm$^3$ were selected to study the friction performance, and the SLM-forming parameters were selected:

scanning speed $v$ = 800 mm/s; layer thickness $t$ = 0.03 mm; hatch space $h$ = 0.08 mm; and power $L$ = 90 W, 190 W.

$$E = \frac{L}{v \cdot t \cdot h} \qquad (1)$$

### 2.1. Porosity and Hardness Test

Polish the upper surface of the SLM-formed sample until there are no scratches visible to the naked eye. Take five images at different positions in the center of the sample, as shown in Figure 2. According to this scheme, the measurement range covers about one-third of the upper surface of the sample. Use Image J software (version 1.7.0) to measure the porosity of 5 images and finally obtain the corresponding porosity by calculating the average value of the porosity. Make 5 indentations in the porosity observation area and find the average value of hardness. The test-loading pressure is 1.96 N (200 gf), and the holding time is 15 s. When testing indentation, try to avoid large pore defects, so as not to affect the measurement results.

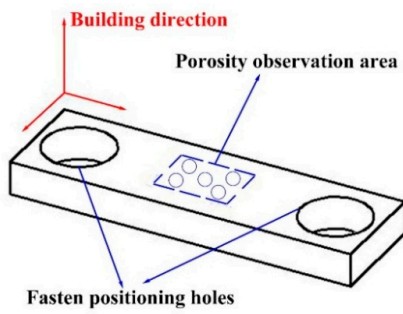

**Figure 2.** Schematic of specimen forming and measurement.

### 2.2. Fretting Wear Test

The fretting wear experiment is carried out with self-made experimental equipment, and the principle is shown in Figure 3. The slider 3 is driven in a reciprocating motion by a screw connected to the stepper motor 1. The number of pulses of the stepper motor enables the precise adjustment of the friction stroke, while the velocity of the movement is regulated by controlling the number of pulses per unit of time. The lower specimen is mounted on fixture 5 and moves in a linear reciprocating motion with workbench 4, while the upper specimen is fixed to a vertical rod on fixture 6 and remains stationary. Weight 8 applies the test load to the specimen. The horizontal elastic beam 9 is deformed by the friction force, which causes the strain gauges 10 to elongate or shorten, and friction force can be measured by means of strain-gauge force transducers.

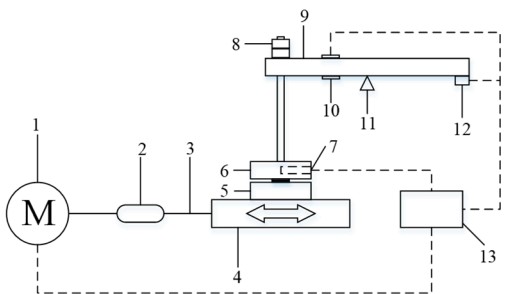

**Figure 3.** Structure diagram of testing machine. 1, Stepper motor; 2, Coupling; 3, Slider; 4, Workbench; 5, Lower specimen fixture; 6, Upper specimen fixture; 7, Temperature sensor; 8, Weight; 9, Horizontal elastic beam; 10, Resistance strain gauge; 11, Pivot; 12, Motion detector; 13, Electrical control system.

The testing machine is shown in Figure 4. In this research, the upper sample is quenched C45E4, the friction head is spherical, the spherical radius is 4.75 mm, the hardness is about 45HRC, and it is fixed. The lower sample is a sample formed by SLM, which

is controlled by a motor to make a reciprocating linear movement and contact the fixed upper sample. The test load is 10 N, the speed is 4 mm/s, the single stroke is 0.6 mm, and the frequency is 2.5 Hz. The friction was carried out under three conditions (dry friction, distilled water, and 3.5% NaCl solution), each set of experiments was repeated at least 3 times, and the duration of each experiment was 60 min. When the water-based solution is used for lubrication and friction, it is necessary to ensure that the contact area is completely enclosed by the liquid. The experiment was carried out at room temperature, and the wear depth and friction coefficient change curve were recorded by computer.

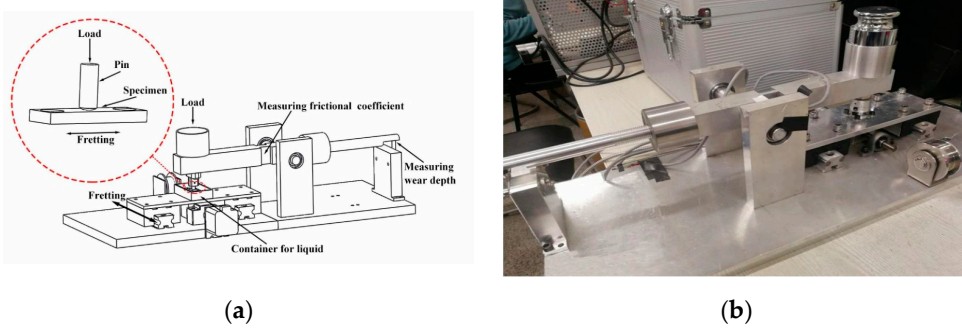

(**a**)    (**b**)

**Figure 4.** Fretting wear testing machine: (**a**) schematic; (**b**) testing machine.

## 3. Results

### 3.1. Porosity Analysis

The pores in SLM are mostly caused by the process. When the energy density is too low, the 316L powder is not completely melted due to insufficient power, resulting in poor bonding between adjacent layers, the extremely irregular shape of the defects, and many irregular pores, such as in Figure 5a. When the energy density is too high, jetting may occur during powder melting. This is because the gas generated by the high energy density is trapped at the bottom of these molten pools to form pores. Therefore, too high energy density will cause the high melting temperature of the molten pool, resulting in high residual stress caused by rapid cooling of the molten pool, and the shape of hole defects is regular and close to circle Shape, as shown in Figure 5b. Figure 6 is the OM image of the sample porosity observation optical microscope. The experimentally measured porosities of the samples with energy densities of 46.88 J/mm$^3$ and 98.96 J/mm$^3$ are 7.66% and 1.00%, respectively. The sample with an energy density of 46.88 J/mm$^3$ has fewer defects than the sample with an energy density of 46.88 J/mm$^3$. The hardness of two samples measured by the microhardness tester is used to calculate the average value. The hardnesses of samples 46.88 J/mm$^3$ and 98.96 J/mm$^3$ are 206.6 HV and 250.6 HV, respectively. Combined with the analysis of porosity, the porosity of the sample with lower energy density is larger, and the corresponding hardness will be smaller.

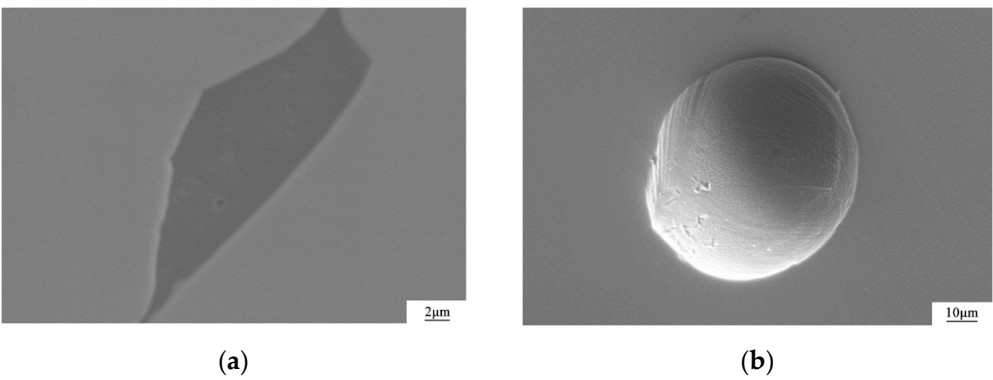

(**a**)    (**b**)

**Figure 5.** Types of defects in SLM-formed 316L: (**a**) irregular defect; (**b**) regular defect.

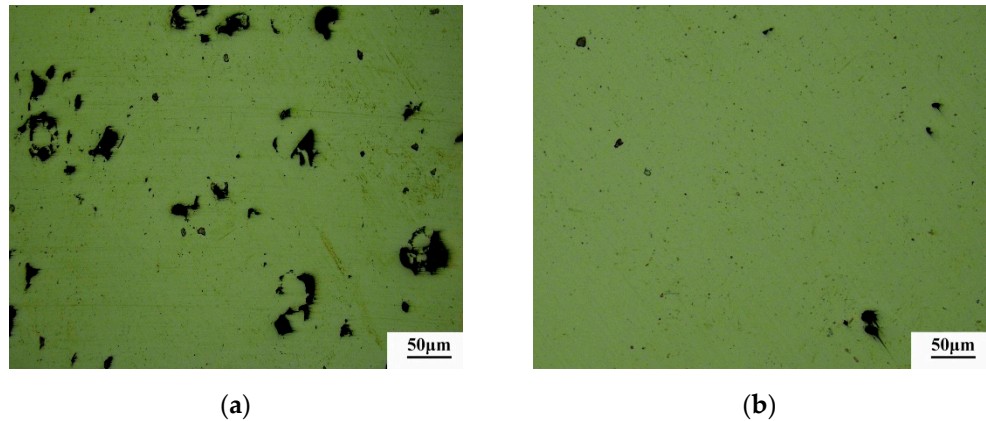

(**a**)        (**b**)

**Figure 6.** OM diagram of porosity observation: (**a**) 46.88 J/mm$^3$; (**b**) 98.96 J/mm$^3$.

### 3.2. Friction Coefficient and Wear Depth

Figure 7a,b are the friction coefficients of samples with energy densities of 46.88 J/mm$^3$ and 98.96 J/mm$^3$ in air, distilled water, and 3.5% NaCl solution. It can be seen in Figure 6 that the friction coefficient of SLM-formed 316L changes with time and can be divided into two main stages. In the first stage, the friction coefficient increases rapidly. In the second stage, the friction coefficient tends to be stable and fluctuates more uniformly around a certain value. When the energy density is 46.88 J/mm$^3$ and 98.96 J/mm$^3$, the friction coefficient during dry friction is stable at about 1.007 and 0.898, respectively. When in the distilled water, the friction coefficient is stable at about 0.567 and 0.534, respectively, and when in the 3.5% NaCl solution, the friction coefficient is stable at about 0.506 and 0.476, respectively. The main reasons for these two stages are as follows: in the first stage, since the surfaces of 316L stainless steel and C45E4 steel start to come into to contact, the movement between the upper and lower pairs increases the frictional resistance and causes adhesion on the contact surface of the two, and the friction coefficient is further increased. In the second stage, the small vibration amplitude of the fretting friction makes it difficult for the wear debris between the two contact surfaces to be discharged, thereby forming a third body. The third body has protective and lubricating effects [21], which can inhibit adhesion. Therefore, when the generation and overflow of debris reach a balance, the friction coefficient changes relatively stable. Furthermore, sudden changes in the friction coefficient can be observed in both stages under dry friction. This is because the material is more prone to peeling and deformation under dry friction, and when large debris peeling occurs, the friction coefficient changes suddenly. As the friction continues, the debris is overflowed or flattened and the friction coefficient re-stabilizes.

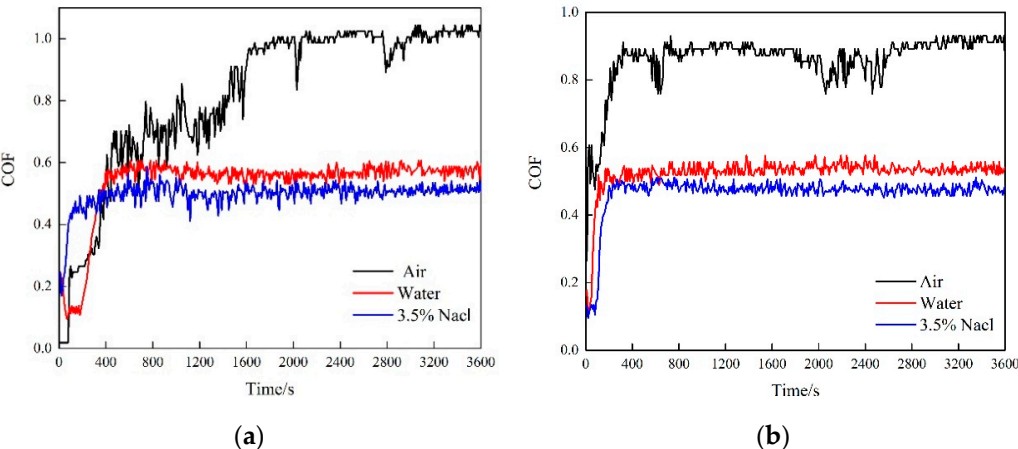

(**a**)        (**b**)

**Figure 7.** The effect of the lubricant on the friction coefficient: (**a**) 46.88 J/mm$^3$; (**b**) 98.96 J/mm$^3$.

Under the two energy densities, the friction in the aqueous solution tends to be more stable and the friction coefficient is smaller than in the dry friction. This is because the liquid plays a role in lubricating and reducing friction [22]. The friction coefficient in the 3.5% NaCl solution is smaller than that in the distilled water. Active elements such as Cl are present in the 3.5% NaCl solution in the form of sodium chloride. Owing to the friction heat generated during the friction process, these elements will react with the element Fe, and an easy-shear friction film containing ferric chloride forms. This prevents direct contact between the friction pairs, and the two-body wear becomes three-body wear, which further reduces the friction coefficient [23–25]. Under the three friction conditions, the sample with an energy density of 98.96 J/mm$^3$ is denser, so it is less prone to wear than the sample with an energy density of 46.88 J/mm$^3$. The closeness of the friction coefficients of the two samples in the aqueous solution is due to the lubrication of the aqueous solution.

Figure 8a is a schematic diagram of the comparison of wear depth. The slope of the wear depth after 30 min is defined as the wear rate (Figure 8b). It can be seen from the figure that the wear depth of the samples is less than 10 μm, the maximum wear depth is generated when the energy density is 46.88 J/mm$^3$ sample dry friction, and the wear depth is about 9.225 μm. The minimum depth is produced in the sample distilled water medium with an energy density of 98.96 J/mm$^3$, and the wear depth is about 2.292 μm. The trend of the wear rate is similar to the wear depth.

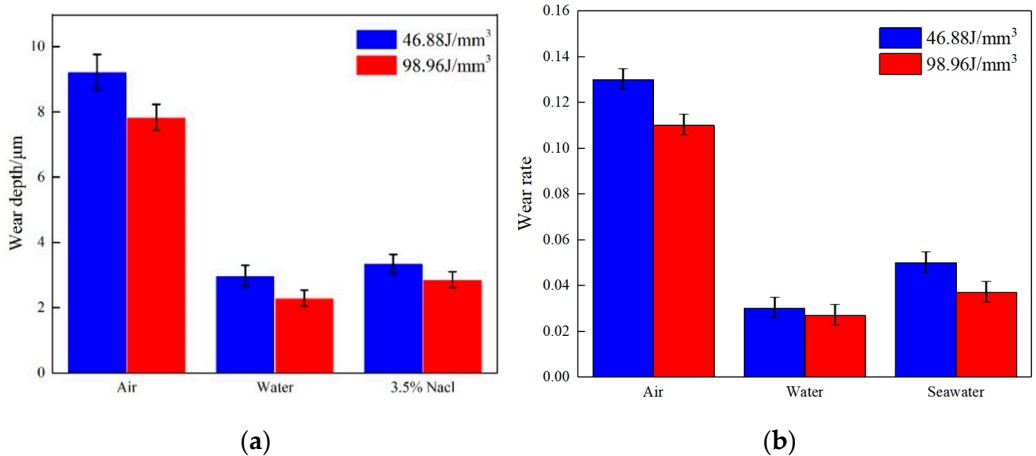

(**a**) (**b**)

**Figure 8.** Wear depth and wear rate. (**a**) Wear depth, (**b**) Wear rate.

Under the three friction conditions, the wear depth and wear rate of the sample with an energy density of 46.88 J/mm$^3$ is greater than that of the sample with an energy density of 98.96 J/mm$^3$. This is due to the fact that the low-energy density sample has more pores and the structure is not dense. When the energy density of the sample is constant, the wear depth and wear rate in the aqueous solution are much smaller than during dry friction. This is mainly because the lubrication effect of the aqueous solution greatly reduces friction and wear. The wear depth and wear rate in the distilled water are lower than those in the 3.5% NaCl solution. In the 3.5% NaCl solution, wear is accompanied by corrosion, so the wear depth and wear rate in the 3.5% NaCl solution are slightly higher than those in the distilled water.

### 3.3. Wear Morphology

Figure 9a is the wear scar and oxygen Energy Dispersive Spectroscopy (EDS) during dry friction. The surface of the abrasion marks was covered with an oxide film, and the surface was severely cracked. In some areas, the oxide film was removed to form wear debris, leaving a dark and rough surface. The oxidation of the sliding wear surface is the result of friction heating in the ambient air. Additionally, in this process, plastic deformation and material peeling are produced, which cause the rough surface of the wear to produce many grooves and cracks. Therefore, the wear mechanism is mainly oxidative wear and

adhesive wear. The wear scar morphology and oxygen EDS in the distilled water are shown in Figure 9b. The wear marks in the distilled water have clear contours, and furrows are formed in the wear marks. Under the cooling and lubricating action of distilled water, the adhesion and peeling of the material surface are inhibited. There are furrows and micro cracks along the fretting direction on the wear scar, and distilled water plays a role in blocking oxygen, so the surface oxidation is inhibited. This indicates that the wear mechanism of 316L stainless steel formed by SLM in the distilled water is mainly abrasive wear, accompanied by fatigue wear. From Figure 9c, it is found that the wear scars in the 3.5% NaCl solution are covered with furrows along the fretting direction and microcracks in parallel and perpendicular fretting directions. Compared with dry friction, 3.5% NaCl solution inhibits the occurrence of oxidation reaction. There is almost no adhesion and material peeling on the wear surface in the 3.5% NaCl solution. The wear surface is mainly composed of furrows and microcracks, so it is in sea water. The wear mechanism is the same as that in the distilled water—mainly abrasive wear accompanied by fatigue wear.

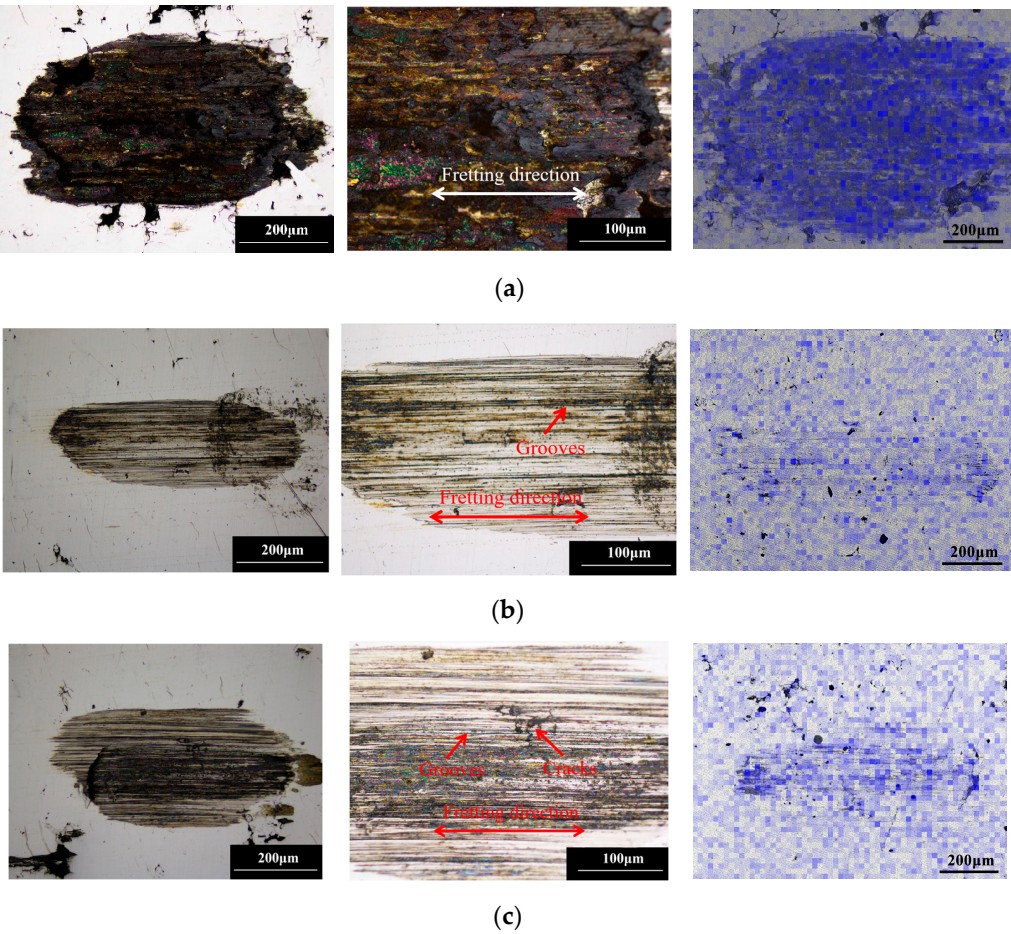

**Figure 9.** OM image of wear scar and EDS image of oxygen element: (**a**) air; (**b**) water; and (**c**) 3.5% NaCl.

## 4. Conclusions

Based on the results obtained, several conclusions can be drawn about the fretting wear characteristics of SLM-formed 316L stainless steel made by the 3D printer in seawater.

(1)  The energy density of SLM will affect the porosity of the 316L sample, thereby affecting the compactness of the sample, and it will further affect the fretting wear characteristics of the material. Within a certain range of energy density, the smaller the energy density, the less dense the material, and the less wear resistant the material is.

(2)  Under the three friction conditions, the friction coefficient of 316L stainless steel formed by SLM first increased and then gradually stabilized. The aqueous solution

will accelerate the stabilization and reduce sudden changes. When the energy density is the same, the friction coefficient is highest in air and lowest in seawater. Compared to distilled water, the easy-shear friction film produced in seawater can further reduce the friction coefficient.

(3)     Under dry-friction conditions, the main wear mechanisms are oxidative wear and adhesive wear. The fretting wear mechanism in the distilled water and 3.5% NaCl solution are the same as abrasive wear accompanied by fatigue wear. The lack of lubrication results in higher wear depth and wear rate than those in aqueous solution. The friction coefficient in the 3.5% NaCl solution is generally smaller than that in the distilled water. However, due to friction and corrosion during this friction process, the wear depth and wear rate are slightly greater those that in the distilled water.

**Author Contributions:** Conceptualization, M.H., P.C. and X.Q.; methodology, M.H. and P.C.; software, M.H.; validation, M.H., P.C. and X.Q.; formal analysis, X.Q.; investigation, M.H.; resources, M.H.; data curation, M.H.; writing—original draft preparation, M.H.; writing—review and editing, P.C.; visualization, X.Q.; supervision, P.C.; project administration, X.Q.; and funding acquisition, X.Q. All authors have read and agreed to the published version of the manuscript.

**Funding:** This research was funded by the National Natural Science Foundation of China (No. 51905032; No. 51975042), and the Science and Technology Innovation Special Fund, Foshan (No. BK20BE024; No. BK21BE018).

**Data Availability Statement:** The processed data required to reproduce these findings cannot be shared at this time because the data also form part of ongoing research.

**Conflicts of Interest:** The authors declare no conflict of interest.

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
