# Peer review of "Fretting Wear Characteristics of SLM-Formed 316L Stainless Steel in Seawater"

_lubricants, doi:10.3390/lubricants11010007_

Round 1

Reviewer 1 Report

The manuscript is well organized and written. This work has certain innovation, but the guiding implications in the introduction should be much clearer.

1. In the introduction, I suggest that some references should be added in order to highlight the values of the present studies.

2. Authors should write out the manufacturer of the metal 3D printer model EP-M100T

3. In the fretting wear test, why did the authors select the applied load of 10N?

4. For the explanation in Figure 6 that the NaCl solution can react with the Fe element to produce a wear layer, provide evidence for this.

5. In the conclusion, the statement " more likely to occur fretting wear " is inaccurate and it is recommended to evaluate its wear resistance.

6. Image labels are not clear enough and the font is too small.

7. The wear rates or the wear loss (gain) of the 316L stainless steel formed by SLM should added in the manuscript.

Reviewer 2 Report

- It is recommended to use its full title instead of abbreviation FML in keywords.

- The number of references given in the article is small. It is recommended to bring more references, especially during the last five years.

- Be sure to include the innovation of the article at the end of the introduction section.

- Provide more specifications of the laser used, including the manufacturer, beam properties and other specifications.

- The caption used for figure 2 should be modified and a sentence that explains the figure should be used.

- Authors should definitely add pictures of the SLM process as well as the fretting wear test that they have done in the article.

- The presented results are complete and appropriate. But scientific and sufficient discussion on the results has not been done. Authors should include a full and comprehensive discussion of the results presented in the paper.

Reviewer 3 Report

As a result of my evaluation for the article titled "Fretting wear characteristics of SLM formed 316L stainless steel in seawater", it can be accepted after the corrections I have mentioned below.

-The reason why the specified experimental parameters were chosen is not expressed enough.

- In the summary part, it should be stated that the production of the materials is carried out with a 3D printer.

-105. line should be fixed as figure.3. In addition, under the title of 2.2.Fretting test machine, how the test machine works should be explained in detail. The experimental parameters chosen according to the capacity of the machine should be examined.

-It should be stated to which material the defects shown in Figure.4 belong.

-The reasons for the sudden changes seen in the curves obtained in Figure.6 should be examined with the support of the literature.

- The EDS analyzes shown in Figure.8 should be presented under the heading 3.3.Wear Morphology.

-Conclusion part is insufficient as it is. It should be rewritten according to the results obtained.

Round 2

Reviewer 2 Report

The manuscript can be accepted in the present form.